# The Functional Traits of Breeding Bird Communities at Traditional Folk Villages in Korea

**Chan Ryul Park** * , **Sohyeon Suk and Sumin Choi**

Urban Forests Research Center, National Institute of Forest Science, 57, Hoegiro, Dongdaemun-gu, Seoul 02455, Korea; m17swannebula@gmail.com (S.S.); ciromi@korea.kr (S.C.)
* Correspondence: maeulsoop@korea.kr; Tel.: +82-2-961-2612

**Abstract:** Interaction between nature and human has formulated unique biodiversity in temperate regions. People have conserved and maintained traditional folk villages (TFVs) dominated with houses made of natural materials, arable land and surrounding elements of landscape. Until now, little attention has been given to understand the traits of breeding birds in TFVs of Korea. The aim of this study was to reveal traits of breeding birds in TFVs and get conservative implications for biodiversity. We selected five TFVs: Hahoe maeul (HA), Wanggok maeul (WG), Nagan maeul (NA), Yangdong maeul (YD), and Hangae maeul (HG). We surveyed breeding birds with line transect methods, and analyzed functional traits (diet type and nest type) of birds in TFVs. Among 60 species recorded, *Passer montanus* (PM), *Streptopelia orientalis* (SO), *Hirundo rustica* (HR), *Pica pica* (PP), *Phoenicuros auroreus* (PA), *Paradoxornis webbiana* (PW), *Microscelis amaurotis* (MA), *Carduelis sinica* (CA) and *Oriolus chinensis* (OC) could be potential breeding birds that prefer diverse habitats of TFVs in Korea. Compared to the breeding birds of rural, urban and forest environments, the diversity of nesting types for birds was high in TFVs. The diverse nest types of breeding birds can be linked with habitat heterogeneity influenced by sustainable interaction between nature and human in TFVs in Korea.

**Keywords:** backyard forest; functional traits; livelihood; nesting guild; pungsu

## 1. Introduction

Since the Rio Declaration, Agenda 21 in Brazil of 1992, international attentions have been given to indigenous people at tropic regions; a number of studies have been conducted to comprehend and suggest the importance of livelihood and biodiversity of local communities. However, there are a couple of well-known or sightseeing sites of traditional folk villages (TFVs) at temperate regions which have intentionally been conserved and maintained by government supports [1–3]. We need to understand the relationship between biodiversity and TFVs at typical monsoon lifestyles of agricultural and forestry cultivating systems in China, Japan and Korea. Biocultural diversity is known to be connected with the interrelationship between dwellers and nature [4], and also with functional roles of habitats, especially from cultural and socio-economic perspectives [5–8]. Tropic regions have high biodiversity in residential areas, while temperate regions have a high biodiversity by products of a long-harmonized relationship between humans and nature. In 1945, German geographer Lautensach [9] commented on the pattern of Korean settlement in comparison with ASEAN (Association of South-East Asian Nations) people. He pointed out the diverse spectrum from slash-and-burn farming to residential agriculture in Korea, contrary to the ASEAN peoples showing the migrating and turning patterns of slash-and-burn farming. Based on his records, Korea showed permanent settled or migrating patterns of slash-and-burn farming at some areas.

However, there are a few villages conserving old lifestyles and residential patterns in Korea. With changing climate and socio-economic conditions, it is very difficult to maintain TFVs on their own finance, so financial supports by central and local government are essential to conserve folk villages and typical residential patterns like thatched houses and tile-roofed houses. Park (2008) [10] suggested that the biodiversity of a rural landscape can be represented by three kinds of interactions like edge effects, landscape complementation and mutual synchronization. However, there has been little survey on the traits of breeding birds in TFVs. This study was conducted to reveal the traits of breeding birds and find a conservation measure in TFVs in Korea.

## 2. Materials and Methods

### 2.1. Study Site

We selected five TFVs based on the settlement of villagers and maintenance of traditional residential circumstances like thatched houses and tile-roofed houses. Maeuls refer to the village in pure Korean language. Among the five villages, three villages (Hahoe, Yangdong and Hangae maeuls) belong to Gyeonsangbukdo province. Wanggok maeul is located at the most northern parts, while Nagan maeul is at the most southern parts of Korean peninsula. The size of Hahoe maeul amounts as 904,821 square meters; the others are below 530,000 square meters. Based on the size of survey areas, we chose survey distance (Figure 1, Table 1).

### 2.2. Bird Survey Method

By using the digital contour maps (1:25,000 or 1:5000) and forest cover maps (1:5000), survey routes were chosen to include all habitats of birds. The name and area of each site were referred to the regional legends of local government. Birds were surveyed three times in each site with the line transect method in the morning from May 1 to July 5 in 2016. To avoid bias from repeated observations of the same individuals, we surveyed birds while walking at the speed of 2 km per hour between 0530 and 0800 on a clear day. Census trails were set up at the length enough to determine the number of species present in each site. All birds seen or heard within 25 m either side of the census trail were identified by song, call, flying type and field mark by eye or with binoculars (8 × 30). All birds seen were recorded and identified by binoculars, song and call, and the number of individuals were counted; the density was calculated as an individual density(ea/km/hr) of each species [11,12].

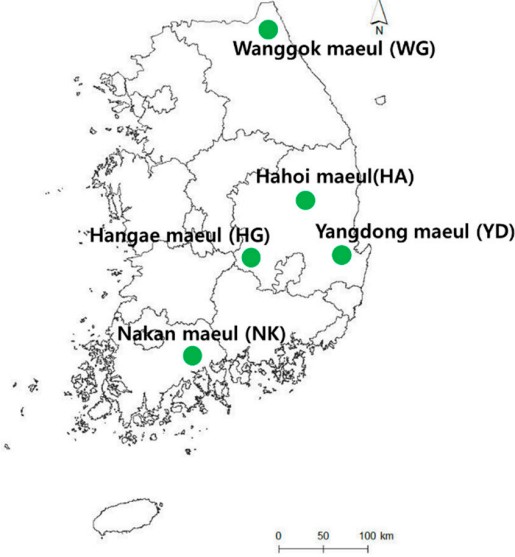

**Figure 1.** Five traditional folk villages (TFVs) were selected based on the villagers' settlements and maintenance of traditional residential circumstances.



### 2.3. Habitat Survey Method

In the viewpoint of bird's habitat, the elements of landscapes at traditional folk village include diverse patches like houses (thatched or tile-roofed), cultivation areas (croplands, paddies and orchards), wetlands (rivers and ponds) and forests. By using the digital contour maps (1:25,000 or 1:5000) and forest cover maps (1:5000), we classified land cover into 13 categories, including paddies, deciduous forests, grasslands, coniferous forests, croplands, rivers, bare lands, residential areas, mixed forests, riparian shrubs, roads, bamboo forests and ponds (Table 2). After classification of land cover, we calculated the percentage value of each cover with the application of ArcMap 9.3 (ESRI, 1999). Habitat diversity and diversity of functional traits were calculated using the Shannon-Wiener diversity index:

$$\mathrm{H}' = -\mathrm{pi}\sum_{i=1}^{n} \ln(pi) \tag{1}$$

**Table 1.** Locations, areas and survey distances at five maeuls.

| Maeuls * (Villages) | Abbreviations | Locations | Area (m$^2$) | Survey Distance (km) |
|---|---|---|---|---|
| Hahoe | HA | Andong City, Gyeongsangbukdo | 904,821 | 2 |
| Wanggok | WG | Goseong County, Gangwondo | 497,560 | 1 |
| Nagan | NA | Suncheon City, Jeollanamdo | 317,548 | 1 |
| Yangdong | YD | Gyeongju City, Gyeongsangbukdo | 392,976 | 1 |
| Hangae | HG | Seongju City, Gyeongsangbukdo | 527,333 | 1 |

* Maeul refers to village in pure Korean language.

**Table 2.** Percentage of each land cover and habitat diversity index at five maeuls.

| Land Cover | HA * | WG * | NA * | YD * | HG * |
|---|---|---|---|---|---|
| Paddies | 32.9 | 5.1 | 27.1 | 1.1 | 3.2 |
| Deciduous forests | 0.4 | 33.8 | 1.8 | 22.8 | 24.2 |
| Grasslands | 10.2 | 4.8 | 27.9 | 34.6 | 9.5 |
| Coniferous forests | 1.9 | 19.8 | 0.0 | 15.2 | 27.3 |
| Croplands | 8.7 | 9.9 | 12.8 | 6.6 | 8.7 |
| Rivers | 17.7 | 0.6 | 1.7 | 0.9 | 0.0 |
| Bare lands | 4.9 | 7.8 | 10.7 | 3.4 | 3.2 |
| Residential areas | 5.1 | 2.0 | 9.2 | 8.2 | 3.1 |
| Mixed forests | 0.0 | 8.8 | 0.0 | 0.0 | 15.5 |
| Riparian shrubs | 13.8 | 0.0 | 0.0 | 0.0 | 0.0 |
| Roads | 4.0 | 3.5 | 6.0 | 7.3 | 2.9 |
| Bamboo forests | 0.0 | 3.8 | 2.2 | 0.0 | 2.4 |
| Ponds | 0.5 | 0.0 | 0.5 | 0.0 | 0.0 |
| Total (m$^2$) | 904,820.7 | 497,560.0 | 317,547.6 | 392,976.3 | 527,332.7 |
| Habitat diversity | 1.9390 | 1.9800 | 1.8522 | 1.7713 | 1.9444 |

* HA: Hahoe, WG: Wanggok, NA: Nagan, YD: Yangdong, HG: Hangae.

### 2.4. Guild Analysis and Functional Guild

We applied functional traits concept to comprehend the nesting resources of breeding bird community, classified into nesting type (hole, canopy, bush (including ground), house and water) and diet type (granivores, insectivores, omnivores, piscivores, predators, scavengers, shorebirds and riverine insectivores) [13,14].

### 2.5. Ordination Analysis and Simple Regression

We conducted non-multidimensional scaling methods to compare the densities of bird communities among five maeuls with the Past 3.13 program [15]. We compared the number of birds and density with habitat diversity with a simple general linear model. We conducted the arcsine transformation of each ratio of land cover for normal distribution of variables.

## 3. Results

### 3.1. Characteristics of Breeding Bird Community at Traditional Folk Villages

We recorded total 60 species of birds at five maeuls, including 20 canopy nesters, 13 hole nesters, 8 bush nesters, 7 water nesters and 5 house nesters. From a viewpoint of diet type, the species were mainly composed of 35 insectivores, 8 predators and 6 piscivores (Appendix A). Among the five regions, Wanggok (WG) maeul showed the highest number of birds, 38 species, and Hangae maeul had the lowest value, 29 species of birds. Hole, bush and canopy nesters were highly observed at Wanggok maeul (WG), while nesting birds near water areas were high at Hahoe maeul (Figure 2a). From the viewpoint of average densities, the values of house nesters were high at Yangdong, Nagan, Wanggok and Hahoe maeul, but low at Hangae maeul (Figure 2b). Diversity of diet types showed the highest value of 1.4967 at Hahoe maeul, and the lowest of 1.2968 at Nagan maeul. Densities of riverine insectivores like wagtails and plovers were high at Hahoe maeul (Figure 2c)

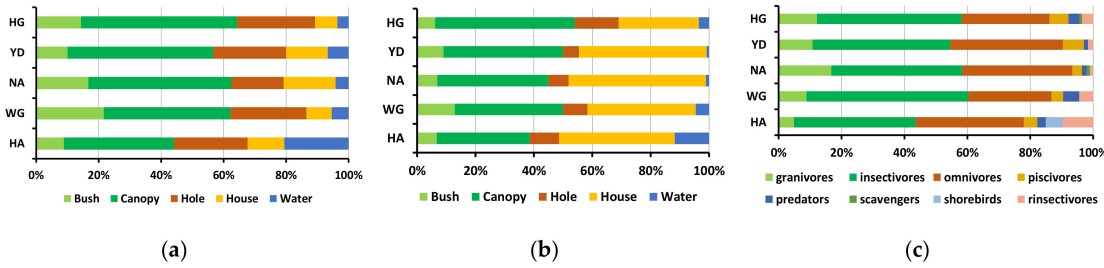

(**a**)                                    (**b**)                                    (**c**)

**Figure 2.** Percentage of nesting type based on the number of species (**a**), average densities (**b**) and diet types based on the average densities (**c**) at five maeuls.

### 3.2. Comparison among Sites and Observed Species

Ordination analysis indicated that Hangae (HG) and Wanggok (WG) maeul showed similarity in the characteristics of bird community; however, the other three sites showed relative unique patterns of bird community (Figure 3). Two sites were composed of high value of forests, 62.4% in WG and 67.0% in HG, so forest birds could dominate the bird community. Hahoe (HA) maeul is located near the river, so riverine birds occurred more highly than at the other sites. Yangdong (YD) maeul showed high percentage of grasslands, and Nakan (NA) maeul a high percentage of bare lands. Thus, among the five TFVs, WG and HG maeuls showed similarity in bird species composition, but the others reflected the characteristics of habitat type for their birds.

Among the observed 60 species, nine species (Passer montanus, Streptopelia orientalis, Hirundo rustica, Pica pica, Phoenicuros auroreus, Paradoxornis webbiana, Microscelis amaurotis, Carduelis sinica and Oriolus chinensis) showed increasing dissimilarity in relation to the other birds which aggregated similarity at the red-dotted circle based on the average density of the five maeuls (Figure 4). Within the red-dotted circle, the birds mostly belong to the forest-dwelling birds. Thus, we could infer that the nine species are related with habitat types of TFVs. Among the nine species, Hirundo rustica and Oriolus chinensis were summer visitors which migrate to Southeast Asia to spend winter, and seven birds were residents.

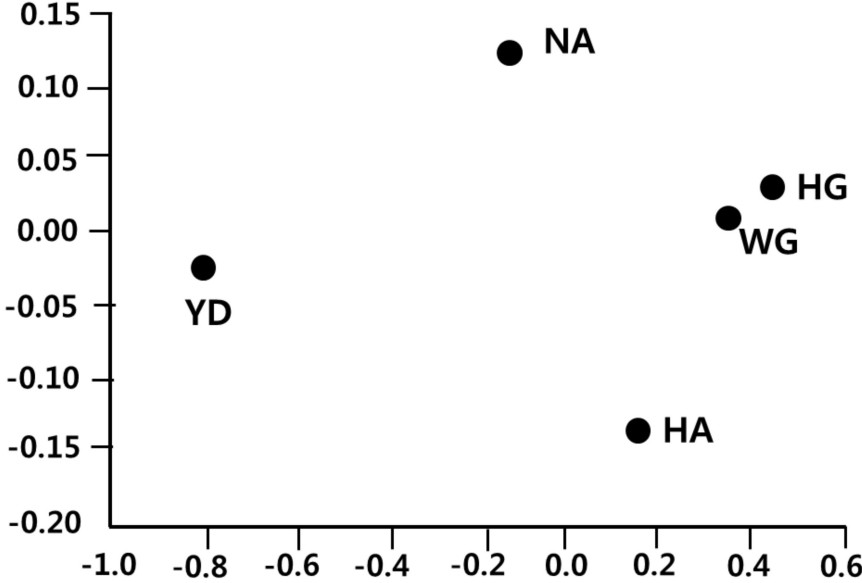

**Figure 3.** Non-multidimensional scaling ordination with the average density of observed birds at five maeuls (YD-Yandong, NA-Nagan, HA-Hahoe, WG-Wangok, HG-Hangae; Stess-0.0, R2-0.9967) by Past, Program V1.35b.

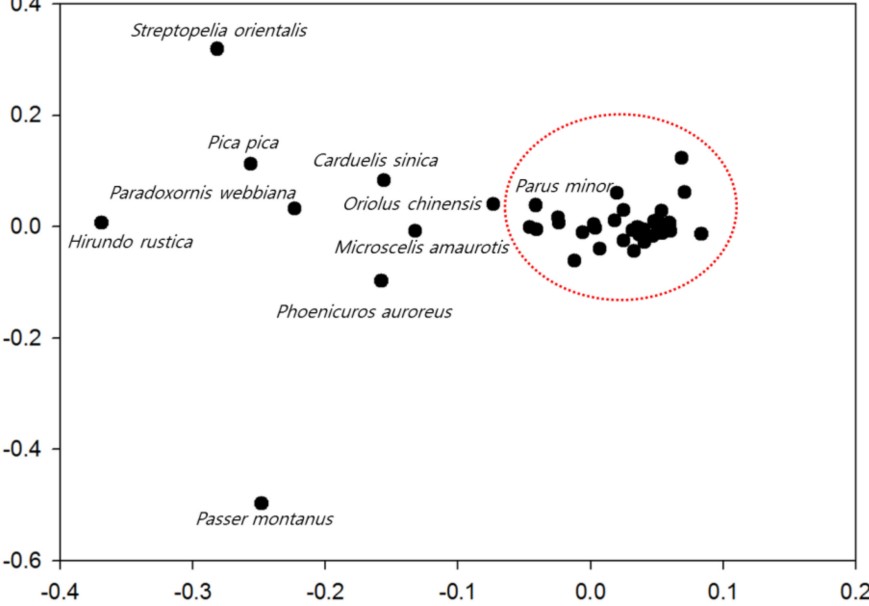

**Figure 4.** Non-multidimensional scaling ordination with the average density of sixty observed birds at five maeuls by Past, Program V1.35b (Red dotted circles include forest-dwelling birds, Stress-0.13, $R^2$-0.7962).

### 3.3. Relationship Habitat Diversity and Number of Birds

At the five traditional folk villages, habitat diversity highly influenced nesting type diversity ($R^2 = 0.74$, Figure 5a), and has a weak relationship with diet type diversity ($R^2 = 0.35$, Figure 5b) and overall number of birds' species ($R^2 = 0.31$, Figure 5c). These results indicate that habitat heterogeneity of traditional rural landscape can provide diverse nesting resources for birds and indirectly affect the functional groups such as insectivores, granivores and predators.

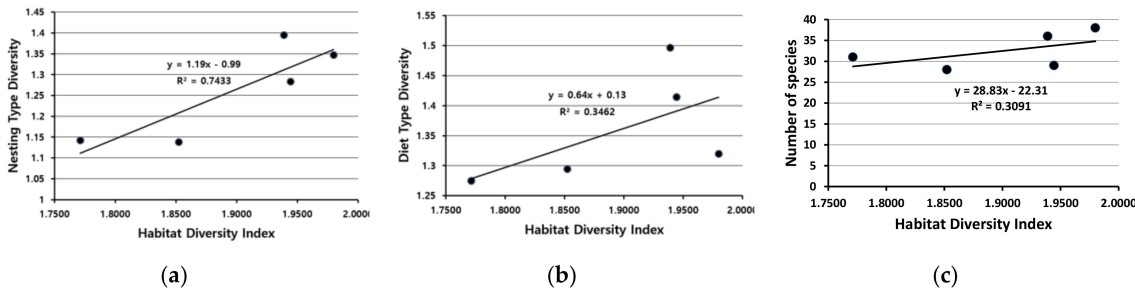

**Figure 5.** Relationships between habitat diversity index and nesting type diversity (**a**), diet type diversity (**b**) and number of species (**c**) at study sites.

## 4. Discussion

### 4.1. The Traits of Breeding Bird Communities at Traditional Folk Villages in Korea

TFVs possess diverse habitats like houses (thatched or tile-roofed), cultivation areas (croplands, paddies and orchards), wetlands (rivers and ponds) and forests for breeding birds. Nine species showed different occurrence patterns with forest-dwelling birds at five maeuls in Korea. This could suggest that nine species (*Passer montanus* (PM), *Streptopelia orientalis* (SO), *Hirundo rustica* (HR), *Pica pica* (PP), *Phoenicuros auroreus* (PA), *Paradoxornia webbiana* (PW), *Microscelis amaurotis* (MA), *Carduelis sinica* (CA) and *Oriolus chinensis* (OC)) can be potential species that prefer diverse habitats of TFVs in Korea. Five out of the nine species were canopy nesters (SO, PP, MA, CS, OC), three were house nesters (HR, PA, PM) and one was a bush nester (PW) according to the nest types. These species reflect the diverse use of nesting resources such as trees, shrubs and traditional thatched house in TFVs. Diversity index of nesting type was higher in the order of TFVs, rural [16], urban [12] and forests [17] (Table 3). Diversity of nesting type could be related with functional traits of breeding bird communities in TFVs.

**Table 3.** Comparison of diversity of nesting types among different systems in Korea.

| Sources | Hole | Canopy | Bush | House | Water | Diversity Index | # of Sites |
|---|---|---|---|---|---|---|---|
| TFVs | 13 | 20 | 8 | 5 | 7 | 1.4880 | 5 |
| Rural [16] | 7 | 10 | 7 | 3 | - | 1.3165 | 43 |
| Urban [12] | 10 | 13 | 8 | - | - | 1.0790 | 2 |
| Forests [17] | 9 | 5 | 5 | - | - | 1.0584 | 6 |

### 4.2. Residential Houses of Nesting Resource for Birds

Among the three house nesters, PM make nests at the multi-porous space of a thatched house with the lining resources of straw of herbs Gramineae and Cyperaceae [18], HR make nests with the muds and straws beneath parts of roof at thatched and tile-roofed house [19], and PA utilize the needles of pines and herbs for lining resources at nests in thatched and tile-roofed house [20]. This represents that the people and birds do coexist and assist breeding success against harsh climate condition and predators. HR was known to prefer the human-dominated house to lessen the predation risk from cuckoos [21]. Thus, residential houses for local people can provide breeding nests for birds in TFVs in Korea. Long-time interaction networks among humans, birds and plants [10] can affect the specific composition of bird community in TFVs in Korea.

### 4.3. Sustainability of Breeding Birds at Traditional Folk Villages in Korea

In Korea, urbanization and industrialization have impacted rural society since the 1970s [22], while TFVs conserve and protect traditional rural landscapes and high biodiversity to this day. Recently TFVs have been cited as pilot villages for low-carbon cities and sustainable urban environments due to COVID-19 and climate change. From this viewpoint, the value and importance of biodiversity

should be evaluated as a new perspective for nature-based solutions like natural healing resources in TFVs. Meanwhile, urban shrinking and climate change could threat the sustainability of nesting & diet types in TFVs in Korea. Recently, biological conservation implication as well as socio-economic policy for stakeholders in rural landscapes could protect and conserve the biodiversity at a specific city and rural community [23–25].

Meanwhile, two summer visitors (HR, OC) do breed in TFVs in Korea, but they migrate to the Southeast Asia to spend winter. To conserve the population of these birds, it is important to ensure wintering grounds and staging grounds at the migration routes in East Asia. Also, it is necessary to assess the habitat of two birds at breeding grounds of Korea and wintering grounds of Southeast Asia. The application of a city biodiversity index [26] would be recommended to evaluate habitats and the biodiversity in TFVs in the Asia region.

Compared to rural, urban and forest habitats, TFVs showed a high diversity of nesting types for breeding birds. The functional traits of breeding birds in TFVs can be related with habitat heterogeneity or simultaneous use by breeding birds at forests, paddies and croplands.

Two cases of the western Amazon [27] and Australian region [28] showed that the importance of biodiversity and livelihoods of indigenous people should be considered before environmental impact assessment of government development policies. From this viewpoint, the relationship between functional traits of breeding birds and the lifestyle of TFVs should be more addressed and interpreted to comprehend the value of TFVs in Korea.

Local & central government have endeavored to conserve TFVs with legislative and economic support to the villagers; however, in-depth excavation of value of TFVs in terms of biodiversity and economic valuation should be executed in the near future. TFVs can provide a harmonized solution of valuable nature and curable nurture in a post-COVID-19 society.

## 5. Conclusions

Villagers in TFVs depend on their livelihoods such as rice production and greens cultivation in agricultural fields, gathering wild edible green and mushroom production in forests. They live in their thatched or roof-tiled houses where swallows and sparrows can safely make breeding nests against predators. Their eco-friendly livelihoods, transcended from old times, allow them to interact with birds, and their harmonized attitudes can enhance biodiversity in TFVs. However, socio-economic change and individualized society can adversely affect the bird diversity as well as local villagers. Research on the interaction between local people and biodiversity should be conducted to sustain TFVs against climate change and COVID-19.

**Author Contributions:** S.S. and C.R.P. collected the field data and wrote the paper; S.S. and C.R.P. collected the field data and analyzed the statistical data on geographic information data; S.C. managed the research project and contributed to the discussion on data; C.R.P. designed the study site and surveyed plans and statistical analyses of all data. All authors have read and agreed to the published version of the manuscript.

**Funding:** This study was funded by National Institute of Forest Science of Korea, grant number NIFOS FE0100-2017-05.

**Conflicts of Interest:** The authors declare no conflict of interest.

# Appendix A

**Table A1.** Densities (ea/km/hr) of bird species included in this study and their functional traits.
RInsectivore = riverine insectivore.

| Scientific Name | Study Sites | | | | | Nesting Type | Diet Type |
|---|---|---|---|---|---|---|---|
| | HA | WG | NA | YD | HG | | |
| *Accipiter gentilis* | 0.0 | 0.0 | 0.3 | 0.0 | 0.0 | Canopy | Predator |
| *Accipiter soloensis* | 0.7 | 0.0 | 0.0 | 0.3 | 0.0 | Canopy | Predator |
| *Actitis hypoleucos* | 0.3 | 0.0 | 0.0 | 0.0 | 0.0 | Water | Shorebird |
| *Aix galericulata* | 2.3 | 0.0 | 0.0 | 1.0 | 0.0 | Hole | RInsectivore |
| *Alcedo atthis* | 0.0 | 0.0 | 0.3 | 0.0 | 0.0 | * | Piscivore |
| *Anas platyrhynchos* | 0.3 | 0.0 | 0.0 | 0.0 | 0.0 | Water | RInsectivore |
| *Anas poeilorhyncha* | 0.7 | 0.0 | 0.0 | 0.7 | 1.3 | Water | RInsectivore |
| *Ardea cinerea* | 1.3 | 0.0 | 1.0 | 3.0 | 1.0 | Canopy | Piscivore |
| *Butorides striatus* | 0.0 | 0.0 | 0.3 | 2.3 | 0.3 | Canopy | Piscivore |
| *Caprimulgus indicus* | 0.3 | 0.0 | 0.0 | 0.7 | 0.0 | Bush | Insectivore |
| *Carduelis sinica* | 2.0 | 0.3 | 0.7 | 8.0 | 0.0 | Canopy | Insectivore |
| *Cettia diphone* | 0.0 | 2.7 | 0.0 | 0.0 | 0.0 | Bush | Insectivore |
| *Charadrius dubius* | 2.0 | 0.0 | 0.0 | 0.0 | 0.0 | Water | Shorebird |
| *Charadrius placidus* | 1.0 | 0.0 | 0.0 | 0.0 | 0.0 | Water | Shorebird |
| *Corvus macrorhynchos* | 0.0 | 0.0 | 0.7 | 0.0 | 0.3 | Canopy | Scavenger |
| *Cuculus canorus* | 1.7 | 1.3 | 1.0 | 2.0 | 0.7 | * | Insectivore |
| *Cuculus micropterus* | 0.7 | 0.0 | 0.3 | 0.0 | 0.0 | * | Insectivore |
| *Cuculus poliocephalus* | 0.0 | 0.0 | 1.0 | 0.0 | 0.0 | * | Insectivore |
| *Cyanopica cyana* | 0.0 | 5.0 | 0.0 | 0.0 | 0.0 | Canopy | Insectivore |
| *Cyanoptila cyanomelana* | 0.0 | 0.0 | 0.0 | 0.0 | 0.7 | Canopy | Insectivore |
| *Dendrcopos major* | 0.0 | 0.7 | 0.0 | 0.7 | 1.7 | Hole | Insectivore |
| *Dendrocopos kizuki* | 0.3 | 0.0 | 0.0 | 0.7 | 0.7 | Hole | Insectivore |
| *Dendrocopos leucotos* | 0.0 | 0.0 | 0.0 | 0.0 | 0.3 | Hole | Insectivore |
| *Dendrocopos* spp. | 0.3 | 0.0 | 0.0 | 0.0 | 0.0 | * | * |
| *Egretta alba* | 1.3 | 0.0 | 0.0 | 2.7 | 0.7 | Canopy | Piscivore |
| *Egretta intermedia* | 0.0 | 0.0 | 0.0 | 0.0 | 0.3 | Canopy | Piscivore |
| *Egretta* spp. | 0.0 | 0.7 | 0.3 | 0.0 | 0.0 | * | * |
| *Eurystomus orientalis* | 1.7 | 0.3 | 1.0 | 2.7 | 1.0 | Canopy | Insectivore |
| *Falco subbuteo* | 0.0 | 0.3 | 0.0 | 0.3 | 0.7 | Canopy | Predator |
| *Falco tinnunculus* | 0.0 | 0.0 | 0.0 | 0.7 | 0.0 | House | Predator |
| *Garrulus glandarius* | 0.0 | 0.0 | 0.0 | 0.3 | 0.0 | Canopy | Omnivore |
| *Halcyon coromanda* | 0.0 | 1.7 | 0.3 | 0.3 | 0.0 | Hole | Piscivore |
| *Hirundo daurica* | 0.3 | 0.0 | 0.0 | 0.0 | 0.0 | House | Insectivore |
| *Hirundo rustica* | 4.7 | 1.7 | 8.0 | 12.7 | 0.0 | House | Insectivore |
| *Lanius bucephalus* | 0.0 | 0.7 | 0.7 | 0.0 | 0.7 | Bush | Predator |
| *Lanius cristatus* | 0.0 | 0.7 | 0.0 | 0.0 | 0.0 | Bush | Predator |
| *Lanius tigrinus* | 0.0 | 0.7 | 0.0 | 0.0 | 0.0 | Bush | Predator |
| *Microscelis amaurotis* | 0.0 | 2.0 | 3.7 | 5.3 | 1.3 | Canopy | Insectivore |
| *Motacilla alba* | 2.0 | 2.0 | 0.7 | 0.3 | 0.0 | Water | RInsectivore |
| *Motacilla grandis* | 0.7 | 0.0 | 0.0 | 0.0 | 0.0 | Water | RInsectivore |
| *Motacilla* spp. | 0.0 | 0.0 | 0.3 | 0.0 | 0.0 | * | * |
| *Oriolus chinensis* | 1.7 | 1.3 | 1.3 | 4.0 | 2.0 | Canopy | Insectivore |
| *Otus scops* | 1.0 | 0.0 | 0.0 | 0.3 | 0.0 | Hole | Predator |
| *Paradoxornis webbiana* | 2.3 | 0.3 | 3.0 | 9.3 | 1.0 | Bush | Insectivore |
| *Parus ater* | 0.0 | 0.0 | 0.0 | 0.0 | 0.3 | Hole | Insectivore |
| *Parus minor* | 0.7 | 0.7 | 1.3 | 2.7 | 2.0 | Hole | Insectivore |
| *Parus palustris* | 0.3 | 0.0 | 0.0 | 0.0 | 0.0 | Hole | Insectivore |
| *Passer montanus* | 14.3 | 9.7 | 19.0 | 36.7 | 6.7 | House | Omnivore |
| *Phasianus colchicus* | 1.3 | 0.7 | 0.7 | 1.0 | 0.3 | Bush | Insectivore |
| *Phoenicuros auroreus* | 4.3 | 5.0 | 2.7 | 3.3 | 3.7 | House | Insectivore |
| *Phylloscopus occipitalis* | 0.0 | 0.0 | 0.0 | 0.0 | 0.3 | Bush | Insectivore |
| *Pica pica* | 7.0 | 2.3 | 4.0 | 7.3 | 4.0 | Canopy | Omnivore |
| *Picus canus* | 0.0 | 0.3 | 0.0 | 0.0 | 0.3 | Hole | Insectivore |

**Table A1.** *Cont.*

| Scientific Name | Study Sites | | | | | Nesting Type | Diet Type |
|---|---|---|---|---|---|---|---|
| | HA | WG | NA | YD | HG | | |
| *Sitta europaea* | 0.0 | 0.0 | 0.0 | 0.0 | 0.3 | Hole | Insectivore |
| *Streptopelia orientalis* | 3.0 | 4.0 | 11.0 | 13.3 | 4.7 | Canopy | Granivore |
| *Sturnus cineraceus* | 1.3 | 0.0 | 2.7 | 1.0 | 0.0 | Hole | Insectivore |
| *Sturnus philippensis* | 0.0 | 0.3 | 0.0 | 0.0 | 0.0 | Hole | Insectivore |
| *Turdus hortulorum* | 0.0 | 0.7 | 0.0 | 0.3 | 0.7 | Canopy | Insectivore |
| *Turdus pallidus* | 0.0 | 0.0 | 0.0 | 0.0 | 0.3 | Canopy | Insectivore |
| *Zoothera dauma* | 0.3 | 0.0 | 0.0 | 0.0 | 0.0 | Canopy | Insectivore |

Study sites: Hahoe maeul (HA), Wanggok maeul (WG), Nagan maeul (NA), Yangdong maeul (YD), and Hangae maeul (HG). * Species were omitted for guild characterization due to the peculiarity of breeding habit or non-breeders.

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
