# Peer review of "The Functional Traits of Breeding Bird Communities at Traditional Folk Villages in Korea"

_sustainability, doi:10.3390/su12229344_

Round 1

Reviewer 1 Report

Characteristics of Avifauna at Traditional Folk Village and Its Implication for Traditional Forest Knowledge in Korea - I think the title is misleading because there is a description of the Avifauna in traditional folk villages but I find no implications for traditional forest knowledge, I also find the term forest knowledge misleading.

In the introduction the authors describe very broad and in general the importance of traditional lifestyle and that it has an implication on biodiversity. I miss the why and how. I also miss some comments why they described bird communities and what does folk villages mean for birds.

In the survey methods the authors describe line and point counts. Had these counts fixed distances or were they open. How did they combine the two methods? In the results they mentioned also density. How did they calculate density and average density? Habitat survey is clear but what is the basis for calculation of percentage. I suggest to order the table of the supplemental material but bird names and not by overall density.

In the results the different villages and habitats were compared descriptive. In the analysis they compared the bird community of the study sites but not the habitat characteristics of the study sites. They found a relation between habitat diversity and birds community diversity, which is, in my opinion, common sense (Are the correlations significant?). They also separated different bird species typical for villages. Is this typical for the study area or is this only a separation because to different habitat types, which means –Did they separate only bird species of settlements or did they really separate bird species typical for traditional folk villages?

From these results they concluded that traditional folk villages are important for bird communities and biodiversity but I miss the comparison with other villages in Korea. There is a qualitative discussion about the importance of different types of houses which are important for some species in the area, off course. I cannot follow the conclusions of the authors about the importance of these villages. I think this study and this analysis does not allow such conclusion without the quantitative comparisons with other types of settlement or areas in Korea. This means at least a more detailed discussion of the results in the light of available literature.

Author Response

Many thanks for your valuable comments.

Reviewer 2 Report

Overview

I have now read and reviewed the manuscript “Characteristics of Avifauna at Traditional Folk Village and Its Implication for Traditional Forest Knowledge in Korea” by Park et al. I would insist the authors throughout the text but also in the title as well to use the term Traditional Folk Villages and not the singular number as you refer to more than one village (in line 40 you also use the plural number to explain TFV). The authors describe functional and ecological traits of avifauna (in my opinion the term “characteristics” which is used in the text is unsuccessful) recorded in five areas. They also calculate habitat diversity and try to relate it with bird diversity. However, it is unclear to me what is the question of the paper based on this analysis and what is the “Traditional Forest Knowledge in Korea” which is included on the title and nowhere throughout the text. I would insist you to strictly reexamine your analyses and your results and try to more accurately rewrite your research question. For example, you describe these villages as an interrelation of nature – human attributes but it is not clear why these TFV need to be sustained based on your results. I have now checked the references used and seem to be updated and suitable for the text except for some cases which I refer to them. The ms needs substantial revision and improvements regarding the English language. Nevertheless, I think that there are a few methodological shortcomings, which I describe below.

1| Bird Survey: How many sites did you sample across the five villages? Is this number scientifically sound for your statistics? You report “Based on the size of survey areas, we chose survey distance” (lines 69-70). How did this emerge? It is also not clear what do you mean with “Census trails were set up at the length enough to determine the number of species present in each site. Did you find all these 60 species in each site you surveyed?  In lines 121-123 you refer also the protection status (protective species, natural monuments, endangered) but this information is not available in the Supplementary material and we do not understand if this information is related to your questions (conservation, sustainability of these villages)?

2| Functional and ecological traits: As far as I have forementioned the characteristics you describe they are more than simple “characteristics” but functional traits which describe important dimensions of ecological bird niche. As such, the term “functional guild” you use is a more general term and has a broader application in animal studies, and in order to delineate guilds scientists use traits. Specifically in my opinion, what you have calculated is the functional trait of “diet type”, the functional trait of “nest type” as well as “foraging location” trait (this is not reported in your methods part, why??). However, its is not clear if you found these data from bibliography or if you recorded them in the field. I check the supplementary file and firstly, you need to explain what are these numbers in the first five columns for each bird species.  Secondly, you need to give us information on these asterisks. Are these asterisks information that you did not manage to record in the field? Because if you have used bibliography you could easily fill in these gaps as this information is broadly available for these bird species. Moreover, I concern about the categories of “diet type” and “foraging type”. In the first case, you refer to this paper of Martin et al. but I got through this paper and I am not sure why you refer to this? Although, I did not find any use of diet type on this paper. You also use shorebirds which is not a diet category but a habitat category because it has to do with what habitats these birds choose to live and not what type of food they feed on. Also, why riverine insectivores category is different from insectivores category? Please explain because I don’t find it correct. Predators are usually carnivores. In the second case, I would like to ask you if you did not find “ground” category at all or you excluded this category from the beginning.

3| Statistical analyses: It is necessary to describe in detail how you performed your analyses. For example, I see in the Figure 1 the ordination plot and it is not clear to me what kind of data you have used to produce that. Are you trying to ordinate the five villages according to the land cover data or on the bird data (e.g. their presence/absence on these villages) or both? Please define. In Lines 158-162 you describe that among 60 species, 9 species which were more separated in the ordination plot are characteristic species for TFV? Please explain what do you mean? The rest 51 species are also found in the TFV. What’s the difference?

4| Relationship of habitat diversity and bird diversity: In this section, I have some concerns on how you have calculated nest type diversity and diet type diversity (the term functional diversity is not a suitable terminology – functional diversity is a broad concept in ecology and is calculated via indices and traits- so please stick on diet type diversity and correct Figure 5b). Please explain in detail how did you calculate nest type diversity and diet type diversity.

5| Discussion: The beginning of the discussion refers to these 9 species for which I have explained my concerns above. So, I cannot understand why these species are typical of TFV. In Lines 213-217 you separate TFV from the other land cover categories. It is like you mean that the TFV is a category of land cover, why?

Minor comments

Title is confusing and hard to read

Line 9: delete “been”

Line 10: delete “been”

Line 11: delete “in making” and use plural for “house”

Use traditional folk villages throughout the text

Line 13: Rewrite your research question because it is not clear

Line 18: analyzed ecological and functional traits of TFV avifauna (diet type, nest type, foraging location)

Line 21: 60 bird species “in” total from which 12 species are legally protective species, seven species are natural monuments and five endangered species

Line 25: functional group diversity? Please rewrite

Lines 27-32: Please focus on your results and the implications of them for conservation and sustainability of TFV and do not repeat words from conclusions

Line 38 and throughout the whole text: people without s NOT peoples

Line 41 instead of “where” use “which”

Line 41: We need…start a new sentence

Line 43: instead of “well related” use “connected”

Line 46: Delete “as well as”

Line 46: tropic regions not tropic regions

Line 48: Delete (1945)

Line 50: Please explain the abbreviation ASEAN

Line 53: “conserve”

Lines 57-59: I am not sure that this sentence fits here

Line 97: After “the”

Line 101: Rewrite according to previous comments

Line 108: We compared bird diversity and density with habitat diversity using a general linear model

Line 181: “were” composed

Line 188: PM often makes

Lines 191-192: This species shows that bird species and people coexist….

Lines 201-202: hard to understand this information in this sentence please rewrite

Line 202: “could” threat

Line 203: instead of & use “and”

Line 204: “implications”

Line 207: So, the international cooperation…. could be considered as an implication

Line 216: impeded?

Line 219: environmental impact assessment

Line 224: could provide

Line 225: Studying biodiversity

Line 237: delete “diverse”. Here the last sentence confuses me. From the one side you use the verb maintain and from the other side you use modify?

Author Response

Many thanks for your detailed explanations and most valuable comments.

Round 2

Reviewer 1 Report

2.2. How did you calculat density?

2.3. To me it is not clear what is the basis for percentage value. The belt where you surveyed the birds or the area of the whole village.

Line 652-654 I miss some literatur when comparing the results of the paper with results from other habitats.

Author Response

We replied to the reviewer comments one by one.

Many thanks for your comments.

Reviewer 2 Report

The ms has been improved. Please find below some minor comments.

Minor comments

Line 18: could be potentially the breeding birds which prefer the…

Lines 470-471: Habitat diversity and diversity of functional traits were calculated using Shannon-Wiener diversity index (formula)

Lines 477-478: here you repeat information, please delete this sentence

Line 481: percentage of what?

Line 559: … showed increased dissimilarity in relation to the other birds which…

Line 561: so we could infer that…

Line 563: ….and seven birds were residents.

Line 565: correct “communitis” to “communities”

Line 589: nine species: delete “of”

Line 592: Five out of nine species were canopy nesters…

Line 593: These species reflect the diverse use

Line 596: with “functional traits” delete “a”

Line 642: “could” threat

Line 644: “could” protect

Author Response

Many thanks for your comments and correction on ms.

We replied your review one by one.